# The MYST Family Histone Acetyltransferase SasC Governs Diverse Biological Processes in *Aspergillus fumigatus*

**DOI:** 10.3390/cells12222642

**Published:** 2023-11-16

**Authors:** Jae-Yoon Kwon, Young-Ho Choi, Min-Woo Lee, Jae-Hyuk Yu, Kwang-Soo Shin

**Affiliations:** 1Department of Microbiology, Graduate School, Daejeon University, Daejeon 34520, Republic of Korea; kjywodbs98@edu.dju.ac.kr (J.-Y.K.); youngho1107@gmail.com (Y.-H.C.); 2Soonchunhyang Institute of Medi-Bio Science, Soonchunhyang University, Cheonan 31151, Republic of Korea; mwlee12@sch.ac.kr; 3Department of Bacteriology, University of Wisconsin-Madison, Madison, WI 53706, USA

**Keywords:** *Aspergillus fumigatus*, histone acetyltransferase, MYST family, SasC, asexual development, stress responses, virulence, transcriptomics

## Abstract

The conserved MYST proteins form the largest family of histone acetyltransferases (HATs) that acetylate lysines within the N-terminal tails of histone, enabling active gene transcription. Here, we have investigated the biological and regulatory functions of the MYST family HAT SasC in the opportunistic human pathogenic fungus *Aspergillus fumigatus* using a series of genetic, biochemical, pathogenic, and transcriptomic analyses. The deletion (Δ) of *sasC* results in a drastically reduced colony growth, asexual development, spore germination, response to stresses, and the fungal virulence. Genome-wide expression analyses have revealed that the Δ*sasC* mutant showed 2402 significant differentially expressed genes: 1147 upregulated and 1255 downregulated. The representative upregulated gene resulting from Δ*sasC* is *hacA*, predicted to encode a bZIP transcription factor, whereas the UV-endonuclease UVE-1 was significantly downregulated by Δ*sasC*. Furthermore, our Western blot analyses suggest that SasC likely catalyzes the acetylation of H3K9, K3K14, and H3K29 in *A*. *fumigatus*. In conclusion, SasC is associated with diverse biological processes and can be a potential target for controlling pathogenic fungi.

## 1. Introduction

Protein acetylation is a conserved evolutionary modification that occurs in eukaryotes and prokaryotes and was first discovered in histones [1,2]. Histone acetyltransferases (HATs) catalyze the acetylation of lysine residues within the N-terminal tails of histone proteins. This modification neutralizes the positive charge of lysines and results in a transcriptionally active chromatin structure, enabling active gene transcription [3].

Based on locations and functions, HATs are classified into two categories: type A HATs are located in the nucleus, and they acetylate nucleosomal histones, while type B HATs are cytoplasmic enzymes that acetylate newly synthesized histones, leading to their transport from the cytoplasm to the nucleus. Type A HATs can be further divided into five families on the basis of the homology of conserved motifs; GNAT (Gcn5-related N-acetyltransferases), MYST (MOZ, YBF2/Sas3, Sas2, and TIP60), p300/CBP (CREB-binding protein), basal transcription factors, and nuclear receptor coactivators [4]. Among them, GNAT, MYST, and the p300/CBP family have been well studied in filamentous fungi [3,5,6,7,8,9].

MYST is one of the largest HAT families; it mediates a diverse variety of biological functions and preferentially acetylates histones H2A, H3, and H4 [5]. The MYST family includes MOZ (monocytic leukemia zinc finger protein), YBF2 (yeast binding factor 2), Sas2 (something about silencing 2), SasC (something about silencing 3), and TIP60 (Tat interactive protein-60) proteins and have a high conserved sequence in the acetyl-CoA binding and zinc finger regions [6]. The most studied MYST histone acetyltransferases in fungi are Esa1 (essential Sas2-related acetyltransferase 1), Sas2, and SasC. SasC is a catalytic subunit of the NuA3 (nucleosomal acetyltransferase of histone H3) complex and responsible for H3 acetylation [10].

In the plant pathogenic and aflatoxigenic fungus *Aspergillus flavus*, MystA (the Sas2 orthologue) and MystB (the SasC orthologue) have opposite functions in sclerotia formation and aflatoxin B1 (AFB1) production, where MystA plays a negative role and MystB plays a positive role [11]. In *Fusarium graminearum*, SasC is indispensable for the acetylation of H3K4, while Gcn5 is essential for the acetylation of H3K9, H3K18, and H3K27. Both are required for DON biosynthesis and pathogenicity [12]. The deletion of *sasC* in *Magnaporthe oryzae* has a significant effect on asexual differentiation, spore germination, and appressorium formation [7]. The deletion of *hat1* (*sasC* homolog) in the insect pathogen *Metarhizium robertsii*, results in a decrease in global H3 acetylation and activation of orphan secondary metabolite genes [13]. Mst2 (SasC orthologue) of another insect pathogen, *Beauveria bassiana,* has shown to regulate global gene transcription through H3K14 acetylation, which enables regulating multiple stress responses and plays an essential role in sustaining the biological control potential against pests [14].

Further evidence of histone acetylation in human fungal pathogens was observed in *Histoplasma capsulatum*, *Cryptococcus neoformans*, and *A*. *fumigatus* [8,15,16]. However, in *A*. *fumigatus*, only the function of the GNAT family HATs has been examined [8,9]. Thus, we examined the functions of the MYST family HAT, SasC, and its influence on development, response to stresses, and pathogenesis. Furthermore, we analyzed the transcriptome of the wild-type (WT) and the deletion mutant to gain insight into the possible roles of SasC.

## 2. Materials and Methods

### 2.1. Strains and Media

All *A. fumigatus* strains used in this study were derivatives of the WT Af293 strain [17]. Fungal strains were grown on yeast extract glucose medium (YG), glucose minimal medium (MMG) or MMG with 0.1% yeast extract (MMY), and appropriate supplements, as described previously [18].

### 2.2. Construction of Mutant Strains

The deletion construct generated by double-joint PCR [19] containing the *A. nidulans* selective marker *AnipyrG* with the 5′ and 3′ flanking regions of the *A*. *fumigatus sasC* gene (AFUA_4g10910) was introduced into the recipient strains [20]. The selective marker was amplified from *A*. *nidulans* FGSC A4 genomic DNA. The complemented strain was also generated via double-joint PCR method [19] with *hygB* as a selective marker. The null mutants and complemented strains were confirmed using diagnostic PCR (using primer pair oligo1497/1498) followed by restriction enzyme digestion (Appendix A). The oligonucleotides used in this study are listed in Appendix A.

### 2.3. Nucleic Acid Manipulation and Analyses

Total RNA was isolated as previously described [21]. Briefly, conidia (5 × 10^5^ conidia/mL) of three strains were inoculated into liquid MMY with appropriate supplements and incubated at 37 °C, 250 rpm. Individual mycelial samples, which were collected at designated time points, were homogenized using a Mini-Beadbeater in the presence of 1 mL of TRIzol^®^ reagent (Invitrogen, Waltham, MA, USA) and 0.3 mL of silica/zirconium beads (BioSpec Products, Bartlesville, OK, USA). The supernatant was mixed with an equal volume of iced isopropanol and centrifuged again. The RNA pellets were washed with 70% ethanol by diethyl pyrocarbonate (DEPC)-treated water and dissolved in the RNase-free water. RNA quality was checked via a spectrophotometer and Bioanalyzer 2100 system (Agilent, Santa Clara, CA, USA). RT-qPCR was performed using One-Step RT-PCR SYBR Mix (Doctor Protein, Seoul, Republic of Korea) and a Rotor-Gene Q real-time PCR system (Qiagen, Hilden, Germany). The amplification of a specific target DNA was verified via melting curve analysis. The expression ratios were normalized to the expression level of the endogenous reference gene *ef1α* [22,23] and calculated via the 2^−ΔΔCq^ method [24]. The expression stability of *ef1α* and efficiencies of PCRs of the target genes were determined, as previously described [25]. The expression stability was determined using BestKeeper index via RefFinder (https://www.heartcare.com.au/reffinder/, accessed on 4 August 2023) [26] and PCR efficiencies of studied genes were 90–102%.

For RNA-seq analyses, total RNA was extracted and submitted to eBiogen, Inc. (Seoul, Korea) for library preparation and sequencing. Construction of the RNA-seq library was performed using QuantSeq 3’ mRNA-Seq Library Prep Kit (Lexogen, Inc., Wien, Austria) according to the manufacturer’s instructions. In brief, each 500 ng total RNA was prepared and an oligo-dT primer containing an Illumina-compatible sequence at its 5′ end was hybridized to the RNA, and reverse transcription was performed. After degradation of the RNA template, second strand synthesis was performed using a random primer containing an Illumina-compatible linker sequence at its 5′ end. The double-stranded library was purified by using magnetic beads and amplified to add the complete adapter sequences required for cluster generation. The finished library was purified from PCR components. High-throughput sequencing was performed as single-end 75 sequencing using NextSeq 500 (Illumina, Inc., San Diego, CA, USA).

### 2.4. Phenotype Analyses

Radial growth was assayed by the inoculation of spores in the center of appropriate media and the measurement of colony diameters. Conidial production was quantified from two inoculation methods. Point-inoculated cultures were used as per the growth area and overlay-inoculated cultures were used on each plate. Conidia were collected using 0.5% Tween 80 solution, filtered through Miracloth (Calbiochem, San Diego, CA, USA), and counted using a hemocytometer (Thermo Fisher Scientific, Waltham, MA, USA). To test for cell wall and oxidative stress, calcofluor white (CFW, 50 μg/mL), Congo red (50 μg/mL), SDS (0.02%), menadione (MD, 50 μM), and H_2_O_2_ (8 mM) were added to the YG media after autoclaving. All chemicals were purchased from Sigma-Aldrich (Burlington, MA, USA). Then, conidia (1 × 10^5^) of relevant strains were inoculated into stressor-treated media and incubated at 37 °C, and colony diameters were measured. UV light tolerance test was carried out as described previously [27]. Briefly, fresh conidia were plated out onto YG plates (100 conidia per plate). The plates were then irradiated immediately with UV using a UV cross-linker and incubated at 37 °C for 48 h. The colony numbers were counted and calculated as a ratio of the untreated control.

### 2.5. Enzyme Assay and Western Blot Analysis

PKA activity was detected with previous method using PepTag^®^ Non-Radioactive cAMP-Dependent Protein Kinase Assay kit (Promega, Madison, WI, USA) [28]. Catalase activity was visualized via negative staining with ferricyanide [29,30]. SOD activity was determined as the inhibition of nitroblue tetrazolium reduction [31]. For Western blotting, histone was extracted with Histone Extraction Kit (Abcam, ab113476, Cambridge, UK), according to manufacturer’s manual. Approximately 50 µg of nuclear protein extract was electrophoresed on a 10% SDS-PAGE gel and subsequently electroblotted to nitrocellulose membranes. Relevant histone modifications were detected with primary antibodies specific to histone H3 (ab1791), H3acK4 (ab176799), H3acK9 (ab177177), H3acK14 (ab203952), H3acK18 (ab40888), and H3acK27 (ab4729) antibodies. Relative intensities of the enzyme activities and Western blot were quantified using the Image J 1.52k software (NIH, Bethesda, MD, USA).

### 2.6. Virulence and Phagocytosis Assay

The virulence assay was conducted, as previously described [25]. Briefly, the CrlOri: CD1 (ICR) (Orient Bio Inc., Seongnam, Republic of Korea) female mice (6–8 weeks old, weighing 30 g) were immunosuppressed by a treatment with cyclophosphamide and cortisone. Mice were intranasally infected with 1 × 10^7^ conidia (10 mice per each fungal strain) suspended in 30 µL of 0.01% Tween 80 in PBS. Then, mice were monitored every 12 h for 8 days after the challenge. Control mice were inoculated with sterile 0.01% Tween 80 in PBS. For histology experiments, the mice were sacrificed at day 3 after conidia infection. Lung tissue sections were stained with periodic acid–Schiff (PAS) and the extent of fungal impact and hyphal growth were compared. Kaplan–Meier survival curves were analyzed using the Log-Rank (Mantel–Cox) test for significance. A phagocytic assay was performed according to a modified method [32,33]. The MH-S cell lines were maintained in RPMI 1640 containing 10% fetal bovine serum (Invitrogen, Waltham, MA, USA) and 50 μM of 2-mercaptoethanol (Sigma, St. Louis, MO, USA). The MH-S cells were adhered to coverslips in 6-well plates at a concentration of 5 × 10^5^ cells/mL for 2 h and subsequently challenged with 1.5 × 10^6^ conidia for 1 h. Unbound conidia were removed by washing with PBS and then incubated for 2 h at 37 °C in an atmosphere of 5% CO_2_. Wells were then washed with PBS and observed using microscopy. The percentage of phagocytosis was assessed.

### 2.7. Bioinformatic Analyses

The amino acid sequences of *A*. *fumigatus* SasC were retrieved from the *A*. *fumigatus* Af293 genomic database (https://fungi.ensembl.org/Aspergillus_fumigatus/Info/Index, accessed on 2 April 2022) and dbHimo (http://hme.riceblast.snu.ac.kr, accessed on 2 April 2022). Amino acid sequences of full length of SasC were subject to the SMART program (http://smart.embl-heidelberg.de, accessed on 4 August 2023) for structural comparison.

### 2.8. Statistics

All experiments performed in triplicate and *p* < 0.05 was considered a significant difference. Data were expressed as mean ± standard error. GraphPad Prism 4 (GraphPad Software, Inc., San Diego, CA, USA) was used for the statistical analyses and graphical presentation of survival curve.

## 3. Results

### 3.1. Summary of A. fumigatus SasC

The estimated 3D structure (Figure 1A) and domain architectures (Figure 1B) of the SasC protein are shown in Figure 1. As shown, the MOZ_SAS domain is located in the central region and the N- and C-terminal segments flank the catalytic core. The predicted *A*. *fumigatus* SasC consists of 1058 amino acids (aa) and has an MOZ_SAS domain (549~724 aa) and the two plant homeodomains (PHD) (203~229 aa, 452~528 aa). The amino acid sequence of the MOZ_SAS domain in *A*. *fumigatus* shows a 96.0~100% identity with those of other *Aspergillus* species. On the other hand, the MOZ-SAS domain of *Saccharomyces cerevisiae* and *Candida albicans* show low similarity, 63.4% and 72.4%, respectively, to that of *A. fumigatus*. A possible active site is glutamate (red in Figure 1B). Glutamate may function to abstract a proton from lysine to promote the nucleophilic attack on the acetyl carbonyl carbon of acetyl-CoA [34,35,36,37].

### 3.2. SasC Is Required for Proper Growth and Development

To investigate the biological functions of SasC, we generated the *sasC* null (Δ) mutant by employing DJ-PCR. As shown in Figure 2A, when grown on MMY and YG media, the Δ*sasC* mutant colony displayed a significantly reduced colony diameter (65%) compared to those of WT and complemented strains (*sasC* C colonies). Moreover, our quantitative analyses of the number of asexual spores (conidia) per growth area and per plate have revealed that the number of conidia of the Δ*sasC* mutant was significantly decreased to about 23% (per growth area) and 77% (per plate) compared to those of WT and complemented strains (Figure 2B). Based on these results, we examined mRNA levels of the central asexual development regulators *abaA*, *brlA*, and *wetA* from the fungal cultures grown in MMY for 3 days. We found that the Δ*sasC* mutant showed highly reduced mRNA levels of all three activators of conidiation (Figure 2C). These results indicate that SasC plays a pivotal role in governing growth and asexual development in *A*. *fumigatus*.

### 3.3. SasC Positively Affects PKA Signaling Pathway and Spore Germination

As the cAMP-dependent protein kinase A (PKA) signaling pathway affects fungal spore germination, growth, and development, we investigate the relationship between SasC and PKA by examining phosphorylation levels of the peptide substrate kemptide, which is specifically recognized and phosphorylated by PKA. The phosphorylated negatively charged kemptide migrates to the anode and the signal is proportional to a higher PKA activity. The Δ*sasC* mutant showed lower PKA activity compared to WT and *sasC* C strains (Figure 3A). To investigate the role of SasC in governing spore germination, we analyzed the kinetics of germ tube emergence in three strains. The germination rate of the Δ*sasC* mutant spores was greatly reduced (Figure 3B). Moreover, as reported by Danion et al. [38], we observed that the conidial germination of all strains tested were asynchronous and that this heterogeneity of germination of the Δ*sasC* mutant conidia was different from that of WT and complemented strains. This might have resulted from modified germination parameters due to the absence of *sasC*.

We also analyzed mRNA levels of PKA signaling components: *acyA* and *pkaC1*. As shown in Figure 3C, *acyA* and *pkaC1* mRNA levels were significantly lower in the Δ*sasC* mutant than in WT and *sasC* C strains. These results indicate that SasC is necessary for proper PKS signaling, which is associated with spore germination, proliferation, and fungal development.

### 3.4. SasC Is Involved in Cell Wall Stress Response

We examined the effects of cell wall stressors on the growth of these strains by exposing them to cell wall stress compounds, calcofluor white (CFW), Congo red (CR), and SDS. The Δ*sasC* mutant exhibited a significantly increased tolerance to Congo red and SDS than WT and *sasC* C strains relative to their control colony grown on solid YG (Figure 4A). These findings suggest that cell wall composition and/or integrity may be affected by the loss of *sasC*. However, we acknowledge that, if there is a limit to the minimum size, the colony reduces upon Congo red and SDS treatment, which would lead to the same colony size of all three strains tested. While unlikely, if this were the case, the Δ*sasC* mutant’s tolerance levels to Congo red and SDS exposure can be reduced. To further test the role of SasC in cell wall biogenesis and integrity, we analyzed mRNA levels of the key chitin biosynthetic genes: *chsB*, *chsE*, and *gfaA*. When induced with Congo red (50 µg/mL, for 6 h), the mRNA levels of *chsB* and *chsE* were significantly decreased in mutant strain compared with WT and *sasC* C strains. In contrast, the mRNA expression level of *gfaA* was markedly increased with the loss of *sasC* (Figure 4B). Previously, it was demonstrated that the mRNA level of *gfaA* had increased upon the treatment of cell wall stressors [39]. These results indicate that the absence of *sasC* affects cell wall biogenesis and cell wall integrity.

### 3.5. SasC Functions in Oxidative Stress Response

To investigate the role of SasC in response to oxidative stress, WT and mutant strains were incubated on YG medium supplemented with oxidative stress agents: 8 mM of H_2_O_2_ and 50 µM of menadione (MD). The Δ*sasC* mutant showed a slightly reduced growth in response to both stress conditions (Figure 5A). To further analyze the role of SasC, we examined the activities of the ROS-detoxifying enzymes. As shown in Figure 5B, activities of all catalases (CatA and Cat1) were significantly decreased in the Δ*sasC* strain. All SOD activities were also considerably reduced upon the loss of *sasC* (Figure 5C). These results indicate that the elevated sensitivity to the external oxidative stressor of the Δ*sasC* mutant might be associated with low catalases and SOD activities. Although the activities of catalases and SODs were dramatically reduced, the expression of other ROS detoxifying enzymes such as glutathione peroxidase and peroxidase was greatly increased with the loss of *sasC* (Appendix A). Therefore, Δ*sasC* mutant may show a relatively weak phenotype in the presence of oxidative stress.

### 3.6. The Role of SasC in Fungal Virulence

In order to assess the pathological roles of SasC, the conidia of three strains were intranasally infected into immunocompromised mice and the pathological outcomes were analyzed by monitoring mouse survival. Mice infected with the Δ*sasC* mutant conidia survived significantly longer than those infected with the WT or *sasC* C conidia (Figure 6A). The absence of *sasC* resulted in a significantly lower pulmonary fungal burden than that of WT and *sasC* C strains (Figure 6B). Further, the interaction of conidia with the murine alveolar macrophage was examined, whereby a significant decrease in phagocytosis was observed in the Δ*sasC* mutant conidia at about 132% of the WT and *sasC* C conidia (Figure 6C). To further understand the basis for the differences in mouse survival, we performed histopathological analysis. The lung tissue sections were prepared from infected mice and stained with periodic acid–Schiff (PAS) to compare the extent of fungal impact and hyphal growth. As shown in Figure 6D, PAS staining revealed a rather small number of fungal cells in sections of the lungs infected with the Δ*sasC* mutant, which were similar to the PBS-treated negative control.

### 3.7. Transcriptome Analysis

To obtain the genome-wide insight into the SasC-mediated processes, we performed Quant-Seq (mRNA-Seq) analyses of ∆*sasC* and WT strains. Of about the 14,000,000 mapped reads, 9833 genes differently expressed (–1 < log_2_ FC < 1) with 2402 genes being significant (at least 2.0-fold, *p* < 0.01), of which 1147 genes were upregulated and 1255 genes were downregulated (Figure 7A). In molecular function gene ontology (GO) categories, “cofactor binding” and “catalytic activity” were upregulated, whereas “signal transducer activity” and “antioxidant activity” were downregulated. The top, upregulated, significant cellular component GO category was “fungal-type cell wall”, whereas “response to stimuli” was the most downregulated GO category in the biological process (Figure 7B). 

As Δ*sasC* mutant showed different responses to external stress compared to WT and *sasC* C strains, we investigated the stress-related transcriptome. Differentially expressed gene responses to stimuli are listed in Appendix A. The highest upregulated gene was predicted to encode an alternative oxidase AlxA (AFUA_2G05060), whereas the highest downregulated gene was predicted to encode a fatty acid oxygenase (AFUA_4G00180) against external stimuli. As UV-endonuclease UVE-1 (AFUA_6G10900) was also significantly downregulated upon the loss of *sasC*, we tried to elucidate the role of SasC in response to UV irradiation. The expression pattern of *uve-1* transcript was first confirmed via qRT-PCR on the same RNA used for mRNA-Seq. As shown in Figure 8A, the mRNA level of *uve-1* was significantly reduced in ∆*sasC* strain (0.14 fold) compared that of WT and *sasC* C strains. The radial growth of the ∆*sasC* mutant was decreased to about 85% of that of WT and *sasC* C strains in response to UV irradiation (Figure 8B). In addition, the tolerance of conidia also severely decreased against UV irradiation upon the loss of *sasC* (Figure 8C).

### 3.8. Potential Targets of SasC

To identify whether histone H3 acetylation levels were changed by the loss of *sasC*, we performed Western blot analyses with specific antibodies against H3acK4, H3acK9, H3acK14, H3acK18, and H3acK27, with antibodies against H3 as a loading control. In the Δ*sasC* mutant, the intensities of signals for all tested antibodies decreased compared with those of WT, except for H3acK4. In particular, the signal intensities for H3acK9, H3acK14, and H3acK27 were remarkably decreased compared to those of WT (Figure 9). These results suggest that SasC catalyzes the acetylation of lysine at the residues 9, 14, and 27 of histone H3.

## 4. Discussion

Histone acetyltransferases (HATs) catalyze the transfer of acetyl groups from acetyl-coenzyme A onto lysine residues of core histones and commonly form a part of complexes [40]. HAT complexes harbor regulatory components that regulate HAT activity and substrate specificity to prevent uncontrolled histone acetylation [41]. HATs are classified into different families, including the GNAT (Gcn5-related N-acetyltransferase) and the MYST (MOZ, YBF2/SASC, SAS2, and TIP60) families [40]. Three MYST families of HATs were identified in human pathogenic fungus *A*. *fumigatus*, EsaA, SasB, and SasC (dbHimo, http://hme.riceblast.snu.ac.kr). Compared to the GNAT family of HATs, roles of the MYST family of HATs remains to be largely understood. In this study, we investigated the roles SASC of *A. fumigatus* as a MYST family HAT in fungal development and pathogenesis.

In budding yeast *S. cerevisiae*, the absence of *sas3* alone does not produce any remarkable phenotypic changes because Gcn5 and Sas3 have overlapping patterns of histone acetylation [42]. However, the deletion of the *sasC* alone significantly impaired vegetative mycelial growth and asexual spore (conidia) production compared to the WT and *sasC* C strains. The deletion mutant not only exhibited diminished radial growth, but also exhibited a reduced production of conidia, indicating that SasC regulated vegetative growth and asexual development. The Δ*sasC* mutant showed a significantly lower PKA activity and germination rate compared to the WT and *sasC* C strains. In addition, the mutant showed the lower mRNA levels of the major components of the PKA signaling pathway: AcyA and PkaC1. These results indicate that SasC may negatively regulate a cAMP-PKA signaling pathway. The deletion of PKA catalytic subunit pkaC1 exhibited a reduction in conidiation, vegetative growth, and pigment formation [43].

Previously, it has been demonstrated that histone acetylation plays only a minor role in the regulation of primary metabolism [44,45,46], but it plays an important role in secondary metabolism [47,48,49,50]. In many fungi, the production of secondary metabolites is associated with acetylation modifications of histone H3. In *A. nidulans*, the activity of several genes for secondary metabolite biosynthesis was associated with the acetylation of histone H3. And the SAGA/ADA complex containing the GcnE was shown to be required for histone H3 acetylation [49]. However, there were no distinct changes in chloroform-extracted secondary metabolites, including gliotoxin (GT), between all tested strains (Appendix A), suggesting that SasC plays minor roles in the regulation of secondary metabolism, which may be due to the absence of the SAGA/ADA complex or the different modes of action in histone acetylation.

The Δ*sasC* mutant showed a significantly lower virulence in a neutropenic murine model than that of WT and *sasC* C strains. This may be associated with the reduction in virulence factors, such as lowered resistance against oxidative stress and upregulated phagocytosis, leading to a lower fungal burden. As a result, through histopathological analysis, we found a very low spore germination and no invasion of hyphae in the lungs of mice infected with the Δ*sasC* mutant. These results indicated that SasC-mediated histone modification is associated with the regulation of pathogenesis in *A. fumigatus*.

Through comparative transcriptomics analyses of WT and the Δ*sasC* mutants, mRNA levels of genes encoding the protein kinase/ribonuclease IreA and bZIP transcription factor HacA were significantly induced in the mutant strain (Appendix A). The protein kinase IreA has a conserved role in response to endoplasmic reticulum stress and is required for the proper localization of the high-affinity iron permease FtrA to the cell membrane. The transcription factor HacA, which is activated by the IreA-mediated removal of the non-canonical intron in the *hacA* mRNA, is dispensable for FtrA localization to the cell membrane and growth under iron-limiting conditions [51]. In addition, HacA plays important roles in the unfolded protein response and is required for the utilization of cellulose in *Neurospora crassa* [52]. Although the expression of HacA significantly increased with the loss of *sasC*, the vegetative growth of the mutant in various carbon sources, including cellulose, was slightly increased (Appendix A). The UV-endonuclease UveA was significantly downregulated with the loss of *sasC* (Appendix A). The *uve-1* is a core gene regulated in response to light and is responsible for tolerance against UV stress for the protection of the mitochondrial genome in *C*. *neoformans* [53]. The results of the mRNA-Seq analysis demonstrate the diversity in cellular processes, especially the utilization of carbon sources and protection against UV, regulated by SasC in *A*. *fumigatus*.

SasC is responsible for H3K9 and H3K14 acetylation and SasC has overlapping patterns of H3 acetylation with Gcn5 in the budding yeast [10]. In numerous fungi, SasC is pivotal for the acetylation of H3K4, H3K9, H3K14, H3K18, and H3K23 [7,11]. SasC of *A*. *fumigatus* may be indispensable for the acetylation of H3K9, K3K14, and H3K29. Substrate specificities have been reported to be mediated by certain subunits from HAT complexes or HAT domains that interact with nucleosomes [54]. In *A. fumigatus*, a PHD finger domain was identified in SasC, with a potential role in substrate specificity or in interaction with regulatory proteins [55].

## 5. Conclusions

In summary, our studies have revealed that the MYST family HAT SasC governs diverse biological processes, such as vegetative growth, asexual sporulation, stress response, secondary metabolites production, and virulence in the human pathogenic fungus *A. fumigatus*. Revealing the crosstalk between histone modifications, their function in effector gene regulation, and the role of transcriptional activators/repressors will help us to further understand the molecular mechanisms linking chromatin- and stage-specific transcriptional changes. Future work aiming to unveil global changes in histone acetylation patterns during infection will shed more light on the contribution of these histone marks to the regulation of the infection machinery.

## Figures and Tables

**Figure 1 cells-12-02642-f001:**
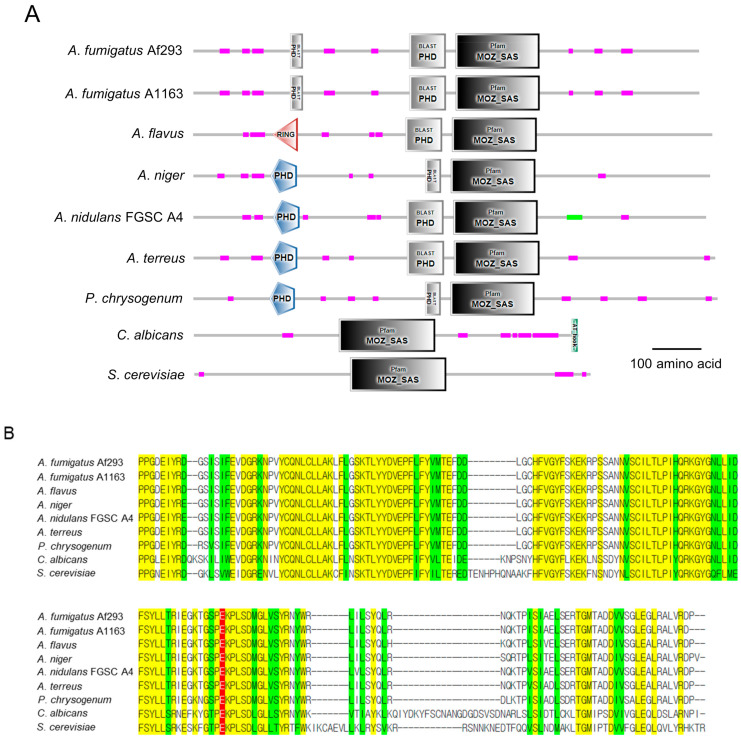
Domain architecture and amino acid alignment of the SasC protein. (**A**) A domain structure of the SasC and SasC orthologs in various fungal species. Domain structures are presented using SMART (http://smart.embl-heidelberg.de, accessed on 4 August 2023). (**B**) Multiple-sequence alignment of the MOZ_SAS domains of SasC and SasC orthologs. Yellow represents conserved residues, while green represents chemically similar residues. Red represents possible active site, respectively. The genomes were used for alignments as follows; AFUA_4g10910, AFUB_067970, AFL2G_01913, NRRL3_07782, ANID_05640, ATEG_03983.1, EN45_098630, orf19.2540, and SC2700752.

**Figure 2 cells-12-02642-f002:**
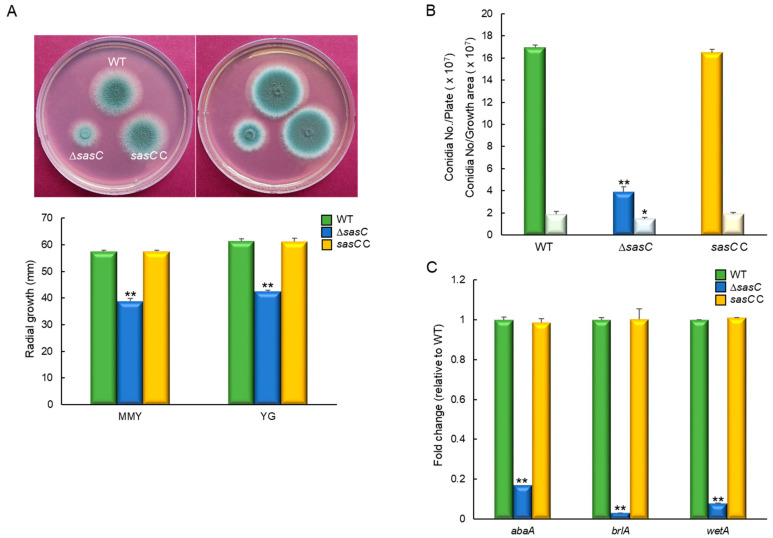
Roles of SasC in vegetative growth and asexual development. (**A**) Colony photographs of WT, ΔsasC, and *sasC* C strains point-inoculated and grown in solid MMY and YG medium. Radial growth of three strains grown on solid media for 3 days determined via colony diameter. (**B**) Conidia numbers produced by each strain per plate (dark color) and per growth area (light color). (**C**) Transcript levels of the key asexual developmental regulators in the mutants and complemented strains relative to those in WT at 3 days determined via quantitative RT-PCR (RT-qPCR). Fungal cultures were grown in MMY, and mRNA levels were normalized to the expression level of the *ef1α* gene. Statistical significance of differences was assessed using Student’s *t*-test: * *p* < 0.05, ** *p* < 0.01.

**Figure 3 cells-12-02642-f003:**
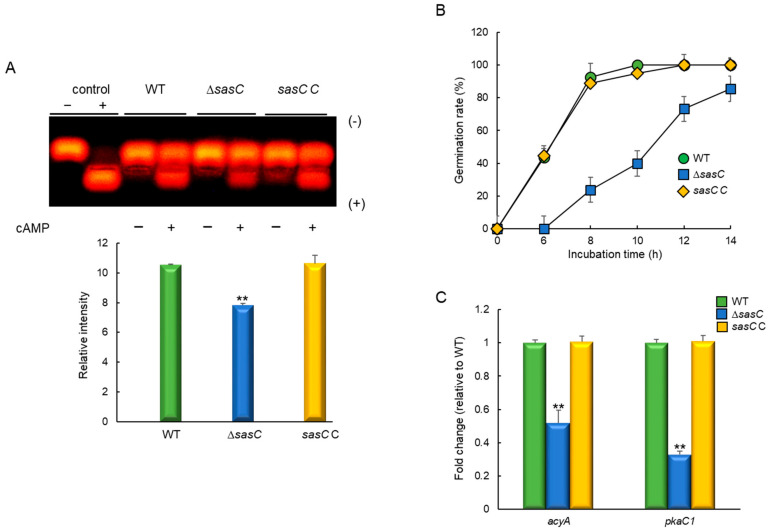
SasC affects the cAMP-PKA signaling pathway and spore germination. SasC affects the cell wall perturbing agents by inducing sensitivity. (**A**) PKA activity levels of three strains as monitored using gel electrophoresis. (**B**) Germination rate of spores. Conidia were inoculated in MMY and incubated at 37° C for 14 h. (**C**) Expression levels of *acyA* and *pkaC1* mRNA in WT, Δ*sasC*, and complemented (*sasC* C) strains analyzed via RT-qPCR. Statistical differences between strains were evaluated via Student’s *t*-test: ** *p* < 0.01.

**Figure 4 cells-12-02642-f004:**
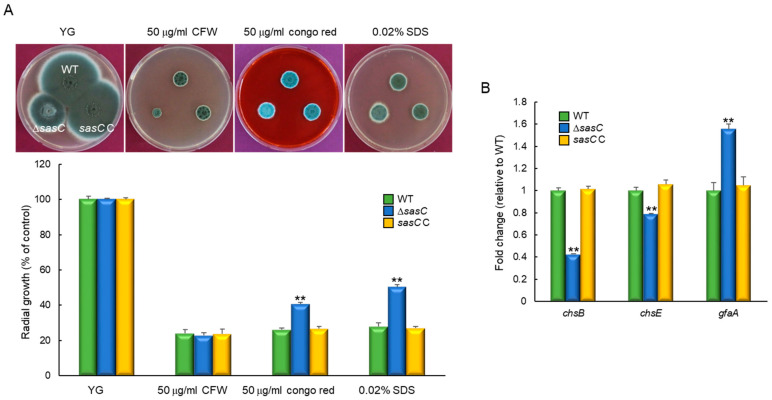
SasC affects sensitivity to cell-wall-damaging agents. (**A**) Colony appearance and radial growth inhibition after inoculation of 1 × 10^5^ conidia on YG-containing cell wall stressors. For the graph, controls were relevant colonies grown on YG. Experiments were performed in triplicate. (**B**) Transcript levels of the key chitin biosynthetic gene *chsB*, *chsE* and *gfaA* in the mutants and complemented strains relative to the corresponding level in the WT strain determined via RT-qPCR. The mRNA levels were normalized to the expression level of the *ef1α* gene. Statistical significance of differences was assessed via Student’s *t*-test: ** *p* < 0.01.

**Figure 5 cells-12-02642-f005:**
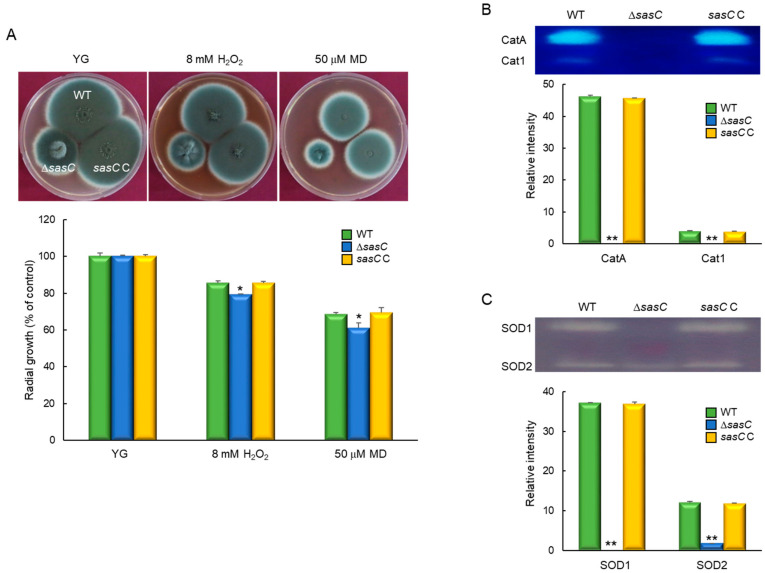
The roles of SasC in response to oxidative stress. (**A**) Colony appearance and radial growth inhibition after inoculation of 1 × 10^5^ conidia on solid YG containing oxidative stressors. For the graph, controls were relevant colonies grown on YG. Experiments were performed in triplicate. (**B**) Catalase activity of the WT and mutant strains. (**C**) SOD activity of the WT and mutant strains. Induction ratios of each enzyme’s activity are shown below. Statistical significance of differences between WT and mutant strains was evaluated using Student’s *t*-test: * *p* < 0.05, ** *p* < 0.01.

**Figure 6 cells-12-02642-f006:**
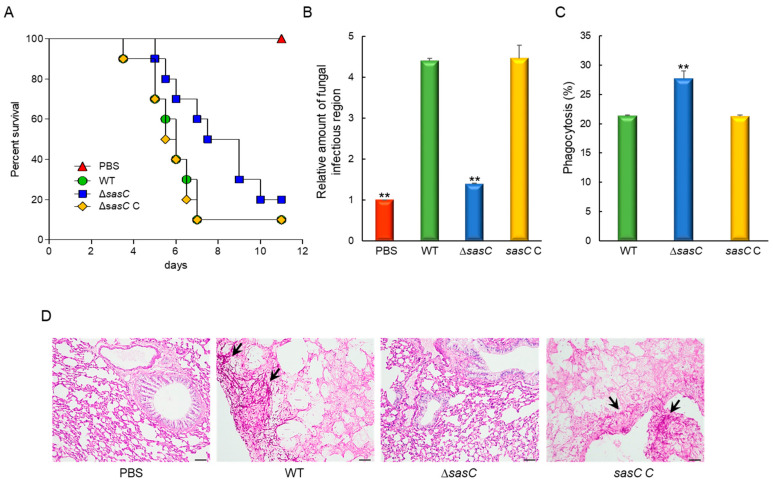
Effects of SasC on the virulence of *A*. *fumigatus*. (**A**) Survival curves of mice intranasally administered with conidia of WT and mutant strains (*n* = 10/group). Kaplan–Meier survival curves were analyzed using the Log-Rank (Mantel–Cox) test for significance (*p* = 0.0507). (**B**) Fungal burden in the lungs of mice infected with WT or mutant strains. (**C**) Phagocytosis of WT and mutant strains. Phagocytosis indicates percentage of macrophages containing one or more ingested conidia (*n* = 20). (**D**) Representative lung sections of mice from different experimental groups stained with periodic acid–Schiff reagent (PAS). Arrows indicate fungal mycelium. Scale bar = 50 μm. Statistical significance of differences between WT and mutant strains was evaluated via Student’s *t*-test: ** *p* < 0.01.

**Figure 7 cells-12-02642-f007:**
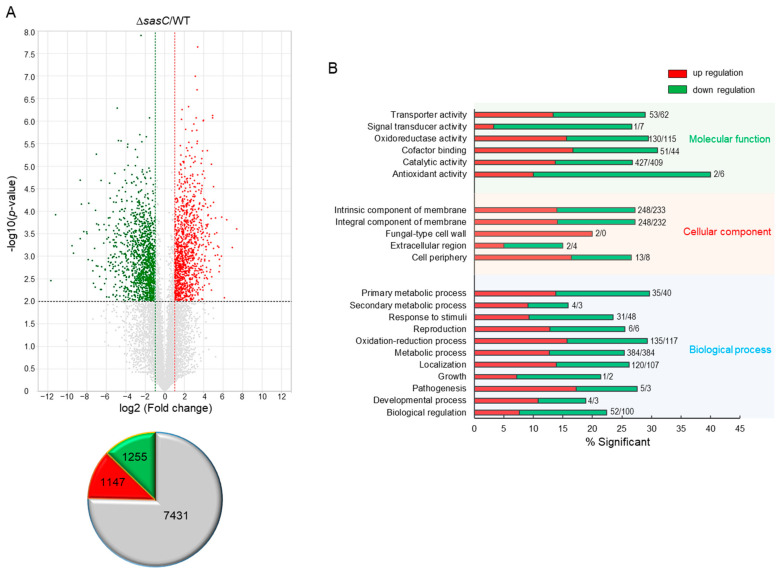
Genome-wide expression analyses of the Δ*sasC* strain. (**A**) Volcano plot showing the fold change (*x*-axis) and *p*-value (*y*-axis) of genes sequenced in Δ*sasC* strain compared to WT. Red and green dots denote up- and downregulated genes, respectively. Insignificantly expressed genes are grey. (**B**) Functional annotation histograms of DEGs in Δ*sasC* strain. The red bars represent genes whose mRNA levels increased in the mutant, whereas the green bars represent those genes whose mRNA levels decreased in the mutant strain. Numbers represent the amounts of significantly regulated genes.

**Figure 8 cells-12-02642-f008:**
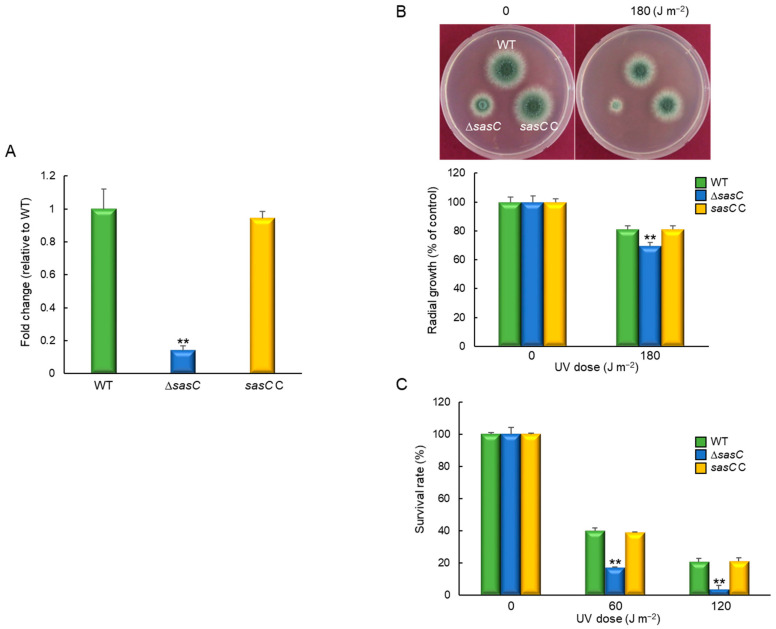
SasC affects tolerance against UV irradiation. (**A**) Expression levels of *uveA* mRNA in WT, Δ*sasC*, and complemented (*sasC* C) strains analyzed via RT-qPCR. (**B**) Colony appearance and radial growth inhibition after inoculation of 1 × 10^5^ conidia on solid YG media. The plates were then irradiated immediately with UV and incubated at 37 °C for 48 h. (**C**) Tolerance of conidia of WT, Δ*sasC*, and *sasC* C strains against UV irradiation. Statistical differences between strains were evaluated using Student’s *t*-test: ** *p* < 0.01.

**Figure 9 cells-12-02642-f009:**
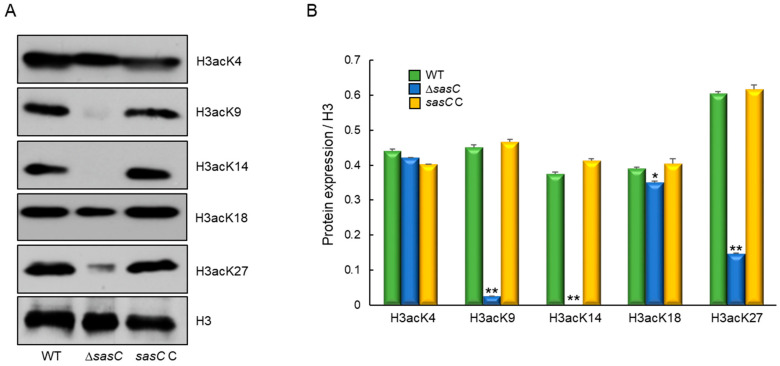
Western blot analysis of histone H3 acetylation levels. (**A**) The anti-acetyl H3K4 (H3acK4), anti-acetyl H3K9 (H3acK9), anti-acetyl H3K14 (H3acK14), anti-acetyl H3K18 (H3acK18), and anti-acetyl H3K27 (H3acK27) antibodies were used for the detection of alterations of acetylation levels. Antibody against H3 was used as a loading reference. (**B**) Quantification of Western blot signals in triplicates. Data were expressed as mean (relative to H3) ± standard error. Statistical significance of differences between WT and mutant strains was evaluated using Student’s *t*-test: * *p* < 0.05, ** *p* < 0.01.

## Data Availability

RNA-Seq data are available from the NCBI Gene Expression Omnibus (GEO) database (GSE166061). The original contributions presented in the study are included in the article/Appendix A; further inquiries can be directed to the corresponding authors.

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
