# Peer review of "The MYST Family Histone Acetyltransferase SasC Governs Diverse Biological Processes in Aspergillus fumigatus"

_cells, 2023, doi:10.3390/cells12222642_

Round 1

Reviewer 1 Report

Comments and Suggestions for Authors

In this paper, the authors characterize the role of the MYST HAT Sas3 in development, transcription and stress resistance in A. fumigatus. Comments and suggested revisions are listed below:

1. Line 169: ‘Possible active site is glutamate’ – Are the authors suggesting that the active site comprises just one amino acid? This should be revised or clarified.

2. Figures 2C, 3C, 4B, 8A - the authors should consider labeling the y axis as ‘Fold change (relative to WT)’ instead of ‘Fold of WT transcripts’

3. Line  224- ‘The Dsas3 mutant exhibited significantly increased resistance to Congo red and SDS than WT and sas3 C strain (Figure 4A).’ –The increased resistance observed here is relative to the control, however, there may be a limit to the minimum size that the colony reduces to upon Congo Red and SDS treatment; the authors should acknowledge this caveat in the text while interpreting this result.

4. In Figure 5, the authors show dramatic changes in catalase and SOD levels in the sas-3 mutant. The oxidative stress sensitivity phenotype, however, is very weak. Can the authors comment on why this is the case, and can they try an alternate oxidative stress paradigm?

5. Can the authors comment on whether any of the transcriptomic changes they observe underlie the stress resistance phenotypes, e.g., the oxidative stress phenotype that they report?

Typos and textual corrections:

1. Line 66, correct to: ‘However, in A.  fumigatus, only the function of GNAT family HATs has been examined.’

2. Line 201, correct to: ‘As the cAMP-dependent protein kinase A (PKA) signaling pathway affects fungal spore germination, growth and development, we investigate the relationship between Sas3 and PKA by examining phosphorylation levels of the peptide substrate kemptide, which is specifically recognized and phosphorylated by PKA.’

3. Line 206, correct to: ‘The sas3 mutant showed lower PKA activity compared to WT and sas3 C strains (Figure 3A)’

4. Line  211, correct to: ‘These results indicate that Sas3 is necessary for proper PKS signaling, which is associated with spore germination, proliferation, and development of the fungus.’

5. Line  245, correct to: ‘To further analyze the role of Sas3, we examined the activities of the ROS detoxifying enzymes.’

Comments on the Quality of English Language

The english should be examined for typos and missing words in sentences. Overall, the english is reasonable.

Reviewer 2 Report

Comments and Suggestions for Authors

The article "The MYST family histone acetyltransferase Sas3 governs diverse biological processes in Aspergillus fumigatus" characterised a MYST family HAT protein in the pathogenic fungus Aspergillus fumigatus. Through exploring phenotypic, molecular and comics experiments the role of this protein is explored. The scientific design is appropriate and conclusions drawn from this are excellent. I do have several comments that should be addressed before this is fit for publication:

Major:

- Many parts of materials and methods include "as previously described". More information is required and this should not be used in general.

- Nomenclature of gene naming is important and should be held to the community standard (https://www.aspergillus.org.uk/genomes/a-proposal-for-the-naming-of-genes-in-aspergillus-species/) This should be changed. In line with this Sas3 should not be used, either choose for SasC or in line with other Aspergillus species MystB can be used. Change this throughout.

- The Alphafold model is not described enough. The scores of the model need to be included. Figure 1 doesn't show what the colours mean. Knowing Alphafold this is the score of the folding which is not good for many of these unfolded loops, likely due to interactions with histones. This model needs to be excluded if the scores are not good enough.

- Figure 3A doesn't show any differences, are there any other examples as this was done in triplicate that are more clear?

Minor:

- L41-42 "well studied" please provide references if claimed.

- L67 grammatically doesn't make sense.

- L76 "as described previously" please provide more detail.

- L89 "as described previously" please provide more detail.

- L95 "as described previously" please provide more detail.

- L96-98 More detail required. What platform sequenced, what library prep etc.

- L106-107 Where were these compounds obtained from and used at what concentration?

- L110 "as described previously" please provide more detail.

- L136 More detail required on histology.

- L148 The aspergillusgenome website has not been updated since 2010, please use FungiDB.

- Figure 1C please include what genomes were used for this alignment. A. fumigatus 1163 should be A1163

- L208-209. This germination data is interesting. It also looks like not only time but heterogeneity is different. Refer to Danion et al 2021 (https://pubmed.ncbi.nlm.nih.gov/33419224/)

- L273-274 Log-rank test and P-value should be included

- L339-340 Not in line with naming. Please see community gene naming rules

- L344 In budding yeast, please mention species.

- L375 Ire1 is IreA in A. fumigatus

- L378 Ftr1 is FtrA

Comments on the Quality of English Language

Minor editing required
